# Synthesis and Evaluation of Chloride-Substituted Ramalin Derivatives for Alzheimer’s Disease Treatment

**DOI:** 10.3390/molecules29153701

**Published:** 2024-08-05

**Authors:** Tai Kyoung Kim, Yongeun Cho, Jaewon Kim, Jeongmi Lee, Ju-Mi Hong, Heewon Cho, Jun-Sik Kim, Yeongyeong Lee, Kyung Hee Kim, Il-Chan Kim, Se Jong Han, Hyuncheol Oh, Dong-Gyu Jo, Joung Han Yim

**Affiliations:** 1Division of Polar Life Sciences, Korea Polar Research Institute, Incheon 21990, Republic of Korea; tkkim@kopri.re.kr (T.K.K.); ashcercle@kopri.re.kr (J.K.); wnal5555@kopri.re.kr (J.-M.H.); kh313@kopri.re.kr (K.H.K.); ickim@kopri.re.kr (I.-C.K.); hansj@kopri.re.kr (S.J.H.); 2School of Pharmacy, Sungkyunkwan University, Suwon 16419, Republic of Korea; okcho9307@naver.com (Y.C.); jungmileedy@naver.com (J.L.); hwcho1012@gmail.com (H.C.); khws123486@nate.com (J.-S.K.); dusrud1129@nate.com (Y.L.); 3Department of Plant Biotechnology, Korea University, Seoul 02841, Republic of Korea; 4Department of Chemistry, Hanseo University, Seosan 31962, Republic of Korea; 5College of Pharmacy, Wonkwang University, Iksan 54538, Republic of Korea; hoh@wcu.ac.kr

**Keywords:** Alzheimer’s disease, Ramalin, derivatives, antioxidant, anti-inflammatory, β-secretase, tau protein

## Abstract

Alzheimer’s disease (AD) is a progressive neurodegenerative disorder marked by the accumulation of amyloid-beta plaques and hyperphosphorylated tau proteins, leading to cognitive decline and neuronal death. However, despite extensive research, there are still no effective treatments for this condition. In this study, a series of chloride-substituted Ramalin derivatives is synthesized to optimize their antioxidant, anti-inflammatory, and their potential to target key pathological features of Alzheimer’s disease. The effect of the chloride position on these properties is investigated, specifically examining the potential of these derivatives to inhibit tau aggregation and beta-site amyloid precursor protein cleaving enzyme 1 (BACE-1) activity. Our findings demonstrate that several derivatives, particularly RA-3Cl, RA-4Cl, RA-26Cl, RA-34Cl, and RA-35Cl, significantly inhibit tau aggregation with inhibition rates of approximately 50%. For BACE-1 inhibition, Ramalin and RA-4Cl also significantly decrease BACE-1 expression in N2a cells by 40% and 38%, respectively, while RA-23Cl and RA-24Cl showed inhibition rates of 30% and 35% in SH-SY5Y cells. These results suggest that chloride-substituted Ramalin derivatives possess promising multifunctional properties for AD treatment, warranting further investigation and optimization for clinical applications.

## 1. Introduction

Alzheimer’s disease (AD) is a progressive neurodegenerative disorder that predominantly occurs in the elderly and is characterized by memory loss, a decline in cognitive function, impaired linguistic abilities, and a reduced capacity to perform daily activities [1,2]. Currently, it affects approximately 50 million people worldwide, and this number is expected to nearly triple by 2050 [3,4,5]. The social cost of AD is significant, affecting individuals, families, and communities. Consequently, developing effective treatment strategies has become a public health priority. However, there is no established definitive treatment for AD, and most therapies primarily focus on alleviating symptoms and slowing disease progression [6]. Currently available treatment for AD include cholinesterase inhibitors (donepezil, rivastigmine, and galantamine) [7] and an NMDA receptor antagonist (memantine) [8]. These treatments provide symptomatic relief but do not halt the progression of the disease. Recent advances also include monoclonal antibodies targeting amyloid-beta (Aβ), such as aducanumab [9] and lecanemab [10], which aim to reduce amyloid plaque buildup, though their long-term efficacy and safety are still under investigation. The etiology of AD is multifactorial and involves genetic, lifestyle, and environmental factors. However, pathological changes within the brain are considered the primary contributors. The two main pathological hallmarks of AD are the extracellular accumulation of amyloid-beta (Aβ) plaques and presence of abnormal intracellular or hyperphosphorylated tau proteins (neurofibrillary tangles) in brain regions associated with memory [11,12]. The accumulation of Aβ disrupts interneuronal communication and induces inflammatory responses, leading to neuronal death [13]. Tau protein, which aids in the stabilization of microtubules and intracellular transport, becomes dysfunctional owing to hyperphosphorylation in AD, leading to abnormal fibril formation and neuronal degeneration [14]. The interplay between Aβ plaque accumulation and the formation of hyperphosphorylated tau proteins involves a complex interaction, where Aβ accumulation promotes tau hyperphosphorylation, and tau hyperphosphorylation further enhances Aβ accumulation, resulting in a vicious cycle that accelerates AD progression [13,15,16]. Beta-site amyloid precursor protein cleaving enzyme 1 (BACE-1) is a major enzyme that promotes the production of Aβ and has been targeted as a principal focus for AD treatment [17]. Inhibiting BACE-1 activity reduces Aβ production and accumulation in the brain, which can reduce cognitive decline [17,18,19]. In addition, derivatives of Ramalin have been shown to inhibit tau activity, as verified by the enzyme-linked immunosorbent assay (ELISA). The ELISA results indicate that Ramalin derivatives significantly inhibit tau protein aggregation. Oxidative stress plays a critical role in the early stages of AD, potentially exacerbating Aβ accumulation. The antioxidant and anti-inflammatory properties of various compounds have been explored, revealing their potential as therapeutic agents against multiple causes of AD. Antioxidants alleviate oxidative stress by providing neuroprotection and delaying the progression [16,20,21]. Similarly, inflammatory responses exacerbate AD, and evidence suggests their involvement even before symptoms appear [22,23]. Targeting anti-inflammatory activity during treatment may offer more preventive than therapeutic benefits against AD.

Previous studies have established the potential of Ramalin and its derivatives as therapeutic agents for AD [24]. To validate this potential, we synthesized a diverse array of Ramalin derivatives. Specifically, we synthesized eight derivatives by varying the position and number of functional groups attached to the phenyl group and investigated their anti-Alzheimer activities. This study focused on substituting the hydroxyl group at the second position of the phenyl group in Ramalin with a chloride functional group. Figure 1 shows the structure of these Ramalin chloride derivatives with their respective synthesis yield indicated as percentages beneath each molecule. The aim was to enhance the antioxidant, anti-inflammatory, and antidementia effects of the Ramalin derivatives by systematically altering the position of the chloride attachment. The introduction of a chloride group was based on its high electronegativity and ability to influence the electronic distribution of the phenyl ring, potentially enhancing the stability and reactivity of the derivatives. Chloride substitution can also increase the lipophilicity of the molecules, improving their interaction with lipid membranes and possibly aiding their passage through the blood–brain barrier (BBB), thereby enhancing bioavailability. In addition, the introduction of a chloride group was intended to optimize the physicochemical properties and biological activities of these derivatives. This foundational research aims to enhance the efficacy of treatments for AD while also exploring therapeutic possibilities for a wide range of neurodegenerative disorders.

## 2. Results

### 2.1. Syntheses of Ramalin Chloride Derivatives

Ramalin chloride derivatives were synthesized using a previously described method [19]. However, this reaction resulted in very low yields (Figure 1). The reaction between Pd and Cl led to the cleavage of the phenyl group. Despite adjustments to a lower temperature (–5 °C) and diluted concentration (approximately 2 mM), the cleavage of the chloride group could not be prevented, and the major product obtained was RA-Ph (70% yield). The desired chloride derivatives were obtained in yields below 5%. When RA-34Cl was synthesized using palladium (Figure 1), the target compound RA-34Cl was obtained in a yield of approximately 5%, whereas the byproducts RA-3Cl and RA-4Cl, in which one chloride was cleaved, were synthesized in yields of 15% and 10%, respectively. The overall yield of the four synthesized derivatives was approximately 80%.

Consequently, the reaction conditions were modified (Figure 2), starting with the protection of glutamic acid with benzyl or carbobenzoxy (Cbz) groups. Then, the reaction was conducted using (*S*)-5-(*tert*-butoxy)-4-((*tert*-butoxycarbonyl)amino)-5-oxopentanoic acid (Boc-glu) as the starting material. This compound was protected with *tert*-butyl and *tert*-butoxy carbonyl (Boc) groups to enable the simultaneous deprotection of the amine and carboxylic acid, respectively, without the use of Pd. A hydrazine coupling reaction was performed according to a previously reported method [24]. The starting material (Boc-glu) was activated at −5 °C using triethylamine (TEA) and ethyl chloroformate (ECF) in dichloromethane (DCM). Specifically, 1.5 equivalents (eq) of TEA were added and stirred for 10 min, followed by the addition of 1.5 eq ECF over the course of 1 h, with the mixture being stirred at the same temperature for 4 h to complete the activation. Then, chloride-substituted hydrazine was added and the mixture was stirred at room temperature (24 °C) for 16 h. The reaction mixture was purified using MPLC to obtain Boc-glu-Hyd. Subsequent deprotection was conducted using 1 M HCl in ethyl acetate (EA) at room temperature (24 °C) with stirring [25]. Finally, the RA-Cl derivatives were obtained as HCl salt crystals. The mixture was filtered and washed with EA to yield the final RA-Cl derivatives.

### 2.2. Blood–Brain Barrier Permeability Evaluation

BBB permeability of the synthesized chloride derivatives was evaluated using swissADME (www.swissadme.ch, accessed on 2 April 2024) (Table 1). The results indicate that the LogP values are unfavorable for BBB permeability. Furthermore, the HCl salt forms of the derivatives showed high ionization in aqueous environments that increased their water solubilities but decreased their lipophilicities, which is unfavorable for BBB crossing. However, owing to their low molecular weights (MW) of 271.70 and 306.15 g/mol, the potential of the derivatives for BBB permeability remains [26]. The total polar surface area (TPSA) values exceeded 100, indicating a disadvantage in BBB permeability, along with a low lipophilicity. Additionally, swissADME marked these derivatives as ‘No’ for the BBB permeant. Despite their low molecular weights, which are advantageous, other properties are largely unfavorable for BBB crossing. Strategies such as prodrug approaches and the addition of nanoparticles or liposomes have been employed to realize favorable properties [27,28].

### 2.3. Antioxidant Effects of Ramalin and Its Chloride Derivatives

The antioxidant activities of the Ramalin derivatives were measured using the 2,2-diphenyl-1-picrylhydrazyl (DPPH) radical scavenging assay. For this analysis, Butylated hydroxyanisole (BHA) was used as the standard reference compound. To assess the antioxidant activities of Ramalin and its chloride derivatives, their radical scavenging activities were evaluated. This was carried out by mixing 150 µL of Ramalin, its derivatives, and BHA solutions at various concentrations (10, 5, 2.5, and 1 mM) in MeOH with 50 µL of 0.1 mM DPPH in MeOH and incubating the mixtures in the dark for 30 min at room temperature. The absorbances were then measured at 540 nm. The chloride-substituted derivatives generally exhibited weaker antioxidant effects than Ramalin. Even in comparison to the previously synthesized derivatives RA-2F and RA-4F, the chloride-substituted derivatives generally had lower antioxidant activities [24]. RA-4Cl showed the highest antioxidant effect with a DPPH IC_50_ value of 7.57 µM. Among the derivatives with two chloride substitutions, RA-34Cl demonstrated the highest antioxidant effect with a DPPH IC_50_ value of 32.84 µM, whereas RA-26Cl had a DPPH IC_50_ value of 135.16 µM, indicating significant activity that was only achieved at high concentrations. The absence of protons at the second or sixth position of the phenyl ring linked to hydrazine in RA-26Cl resulted in considerably reduced antioxidant activity. This result suggests that the antioxidant effect of Ramalin is influenced by electronic resonance involving hydrazine and the protons at position 2 or 6 of the phenyl ring. This finding is consistent with previous results for the synthesized derivative RA-PF [24].

### 2.4. Anti-Inflammatory Effects of Ramalin and Its Chloride Derivatives

In addition to Aβ plaques and neurofibrillary tau tangles, neuroinflammation is also involved in the pathogenesis of AD. Previous studies have shown that the NLRP3 inflammasome is significantly upregulated and activated in AD patients [29,30]. Cyclooxygenase-2 (COX-2) is also hypothesized to be involved in brain inflammation. Although the increase in COX-2 expression in individuals with AD is debated, epidemiological studies have revealed that people who have used nonsteroidal anti-inflammatory drugs for a long time have a lower risk of AD [31,32].

We tested the anti-inflammatory effects of Ramalin and its chloride derivatives on lipopolysaccaride (LPS)-induced inflammation using the murine microglial cell line BV-2 (Figure 2A–C). The expression of NLRP3 and COX-2 was measured in LPS-stimulated cells, centrifuging the lysates, and performing Western blot analysis. Equal amounts of protein were separated by SDS-PAGE, transferred to PVDF membranes, and incubated with primary antibodies against NLRP3 and COX-2. Protein bands were detected using enhanced chemiluminescence and quantified densitometrically. Unexpectedly, Ramalin and its chloride derivatives did not exhibit significant anti-inflammatory effects. Western blot analysis indicated that Ramalin and several chloride derivatives, including RA-2Cl, RA-3Cl, RA-4Cl, RA-23Cl, RA-24Cl, RA-26Cl, RA-34Cl, and RA-35Cl, exhibited low inhibitory effects on the expression of NLRP3 and COX-2. Although the inhibitory effect of Ramalin on NLRP3 and COX-2 was not observed, its inhibitory effect on VCAM-1 has been reported [27]. In addition, previous studies have confirmed its concentration-dependent inhibition of nitrogen oxide (NO) [18]. Furthermore, we found the cytotoxicity and NO inhibitory activities of Ramalin and its chloride derivatives to be similar. Nitrite accumulation was used as an indicator of NO production, and nitrite levels were determined using Griess reagent. BV-2 cells were treated with Ramalin and its derivatives followed by LPS stimulation, and nitrite concentration was measured by the absorbance at 540 nm. The chloride derivatives showed no cytotoxicity, except for RA-2Cl and RA-4Cl (Figure 3). In addition, RA-2Cl, RA-3Cl, RA-4Cl, RA-34Cl, and RA-35Cl caused concentration-dependent reductions in the NO levels (Figure 3). At a concentration of 40 µM, the NO inhibition activities were measured as follows: RA-2Cl at 39.4%, RA-3Cl at 37.15%, RA-4Cl at 46.3%, RA-34Cl at 41.5%, and RA-35Cl at 35.0%. Among these, RA-34Cl exhibited the highest NO inhibitory activity without cytotoxicity. In conclusion, although the inhibitory effects on NLRP3 and COX-2 are minimal or absent, the observed NO inhibitory activity suggests a potential anti-inflammatory effect. Further research is required to determine whether NO inhibition translates into an anti-inflammatory effect.

### 2.5. BACE-1 Inhibitory Effects of Ramalin and Its Chloride Derivatives

BACE-1 is a type I membrane protein and aspartyl protease. It is responsible for the production of Aβ peptide in conjunction with the γ-secretase complex. Aβ is a byproduct of the consecutive proteolytic cleavage of the amyloid precursor protein (APP) by BACE-1 and γ-secretase. The cleavage of APP by BACE-1 is the rate-limiting step in Aβ production; hence, BACE-1 is regarded as a major target for AD treatments. In this study, we evaluated the effects of Ramalin and its chloride derivatives on the regulation of BACE-1 expression using the murine and human neuroblastoma cell lines N2a and SH-SY5Y, respectively. Ramalin and RA-4Cl significantly decreased BACE-1 expression in N2a cells (Figure 4A,B). These compounds had no significant effect on the regulation of BACE-1 in SH-SY5Y cells. Two chloride derivatives, namely RA-23Cl and RA-24Cl, decreased BACE-1 expression by approximately 30% and 40%, respectively, compared with that of the control. However, these decreases were not statistically significant (Figure 4C,D).

### 2.6. Tau Inhibitory Effects of Ramalin and Its Chloride Derivatives

A tau ELISA kit was used to evaluate the efficacy of the chloride derivatives in inhibiting tau aggregation. The results are summarized in Figure 5, showing the relative tau inhibition rates for each derivative compared with those of the control and Ramalin. Both the inhibitor and Ramalin inhibited tau aggregation by 39.6% and 35.8%. In contrast, the chloride derivatives demonstrated higher levels of inhibition than the parent compound, Ramalin. Except for RA-2Cl (32.2%), all derivatives showed inhibition rates exceeding 40% (RA-23Cl at 41.0%, RA-24Cl at 48.1%), with RA-3Cl (50.4%), RA-4Cl (50.6%), RA-26Cl (50.8%), RA-34Cl (54.9%), and RA-35Cl (54.9%) showing 50% inhibition. Specifically, RA-34Cl and RA-35Cl exhibited inhibition activities of 54.9%. Interestingly, the activity varied slightly depending on the position of the chloride. Derivatives with chlorides at positions 3 and 4 exhibited higher tau inhibitory activities than those with a chloride at position 2. Among these, the chloride at position 3 appeared to be particularly associated with tau inhibitory activity. In conclusion, the tested chloride derivatives showed equal or higher inhibitory activities than Ramalin, indicating their potential for tau inhibition. Notably, RA-34Cl and RA-35Cl exhibited the highest tau inhibitory activity, with RA-34Cl, RA-35Cl, and RA-24Cl also demonstrating BACE-1 inhibitory activity. All data were expressed as the mean ± standard deviation of three independent experiments. The statistical significance was set at * *p* < 0.05, ** *p* < 0.01, *** *p* < 0.005.

## 3. Discussion

In this study, we synthesized a series of chloride derivatives of Ramalin and evaluated their potential as inhibitors of tau aggregation and BACE-1 activity, as well as their overall efficacy in mitigating the pathological features of AD. AD is characterized by the accumulation of Aβ plaques and neurofibrillary tangles composed of hyperphosphorylated tau proteins, which contribute to neuronal dysfunction and cognitive decline. Considering the multifaceted nature of AD, therapeutic agents that can simultaneously target multiple aspects are highly desirable.

The synthesis of Ramalin chloride derivatives presents significant challenges, primarily because of the tendency of the chloride group to cleave during the reaction. Despite the optimization of the reaction conditions, the yield is low, necessitating a modified synthetic approach. The final derivatives were synthesized by replacing the glutamic acid protecting groups of the precursor, namely benzyl or Cbz, with *tert*-butyl or Boc protecting groups, respectively. This approach facilitated the simultaneous deprotection of the amine and carboxylic acid groups without the use of Pd, thus overcoming the issue of chloride cleavage. Deprotection of the Boc group significantly enhanced the water solubility of the derivative owing to the formation of a HCl salt. While Ramalin and its derivatives generally exhibited high water solubilities, certain derivatives, such as RA-PF, were nearly insoluble in water. This issue was overcome by modifying these compounds into their HCl salt forms to improve their solubilities.

Evaluation of BBB permeability using swissADME indicated that the chloride derivatives had unfavorable properties for BBB crossing. A high TPSA and low lipophilicity were significant barriers, despite their low molecular weights. These findings suggest that while the derivatives possess potential owing to their molecular size, modifications such as prodrug strategies, nanoparticle addition, or liposomal delivery systems are necessary to enhance their BBB permeabilities.

The antioxidant properties of the derivatives were assessed using the DPPH assay. The results indicate that while the parent compound, Ramalin, exhibited strong antioxidant activity, the efficacy of the chloride derivatives varied. Notably, the derivatives with chloride substitutions at positions 2 and 4 showed enhanced antioxidant activities compared with those with a substitution at position 3. This differential activity underscores the importance of the position of the functional groups in influencing the biological activity of these compounds.

Neuroinflammation plays a critical role in the pathogenesis of AD. The NLRP3 inflammasome and COX-2 are key mediators of inflammatory response. Our results indicated that Ramalin chloride derivatives did not significantly suppress the expression of these inflammatory markers. However, RA-3Cl and RA-4Cl caused a slight reduction in expression compared with the other chloride derivatives. In conclusion, the anti-inflammatory effects of chloride derivatives were minimal; it is unlikely that they can effectively inhibit AD progression through anti-inflammatory mechanisms.

Our study demonstrated that 10 µM of Ramalin and RA-4Cl significantly decreased BACE-1 expression in N2a cells. This is a crucial finding as it highlights the dual functionality of these derivatives in targeting both major pathological hallmarks of AD, amyloid plaques, and tau tangles. Interestingly, the BACE-1 expression in SH-SY5Y cells decreased by approximately 30% and 40% after the addition of the chloride derivatives RA-23Cl and RA-24Cl, respectively, although these reductions were not statistically significant. This suggests that while the primary focus may be on tau inhibition, some derivatives also possess secondary mechanisms that can further mitigate AD pathology by reducing Aβ production.

The tau ELISA results clearly indicated that all tested chloride derivatives, except RA-2Cl, exhibited greater than 40% inhibition of tau aggregation, with several derivatives, including RA-3Cl, RA-4Cl, RA-26Cl, RA-34Cl, and RA-35Cl, achieving nearly 50% inhibition. This finding is particularly significant because tau hyperphosphorylation and its subsequent aggregation into neurofibrillary tangles are key pathological features of AD. By effectively inhibiting tau aggregation, these derivatives can potentially halt or even reverse the progression of neurodegeneration in patients with AD. The data also revealed that the position of chloride substitution significantly affects inhibitory activity. Derivatives with chlorides at the third and fourth positions displayed higher tau inhibition than those with a chloride at the second position. Among these, the third-position chloride derivatives, namely RA-3Cl, RA-34Cl, and RA-35Cl, showed the most potent activities, suggesting that electronic and steric effects at this position favor interactions with tau or its aggregates.

Comparison of the efficacy of these derivatives to that of the parent compound, Ramalin, underscores the importance of structural modifications. Although Ramalin itself demonstrated 35.86% inhibition of tau aggregation, the introduction of chloride groups, particularly at positions 3 and 4, significantly enhanced the inhibitory activity. This enhancement suggests that further exploration of these modifications may yield more potent inhibitors. In addition, the low BBB permeability of these derivatives, as indicated by their unfavorable LogP values and TPSAs exceeding 100, poses a challenge to their development as central nervous system-active drugs. Strategies such as prodrug approaches, nanoparticle delivery systems, and liposomal formulations can potentially improve BBB permeability and overall bioavailability.

Given the promising results of RA-24Cl, RA-34Cl, and RA-35Cl for both tau and BACE-1 inhibition, further studies are warranted to optimize these derivatives for clinical applications. Potential strategies for clinical optimization include conducting structure–activity relationship (SAR) studies to enhance tau aggregation and BACE-1 inhibitory activities, utilizing prodrug approaches to improve BBB permeability, employing nanoparticle or liposomal delivery systems for targeted delivery to the brain, and performing in vivo studies to determine therapeutic potential, optimal dosage, and safety profiles. Chloride-substituted Ramalin derivatives differ from current AD treatments like cholinesterase inhibitors and NMDA receptor antagonists by targeting both tau aggregation and BACE-1 activity, potentially slowing or reversing disease progression. In conclusion, our findings suggest that chloride-substituted derivatives of Ramalin are promising multifunctional agents against AD. The ability to inhibit both tau aggregation and BACE-1 activity renders these derivatives promising candidates for further development in the pursuit of effective treatments for AD.

## 4. Materials and Methods

### 4.1. General Experimental Information

All solvents and reagents were obtained from commercial suppliers (Merck, Darmstadt, Germany) or TCI (Tokyo, Japan) and used without further purification. All glassware were thoroughly dried in a drying oven (60 °C) or flamed and cooled down under a stream of dry argon immediately prior to use. Filters were obtained from a commercial supplier, namely GE healthcare (GF/F, 0.7 µm, Whatman, UK). All reactions were performed under an inert argon atmosphere. Solvents and liquid reagents were transferred to a syringe prior to use. Organic extracts were dried over Na_2_SO_4_ and concentrated under reduced pressure in a rotary evaporator (Eyela, Tokyo, Japan). Residual solvent was removed under high vacuum (Vacuubrand RZ 2.5, Wertheim, Germany, 1 × 10^−2^ mbar). Purification was performed using a Yamazen Smart Flash EPCLC AI-580S (Yamazen, Osaka, Japan) medium-pressure liquid chromatography (MPLC) system. Accurate mass spectra were obtained using an AB Sciex Triple TOF 4600 (Framingham, MA, USA) instrument with the interface in the direct injection mode. Nuclear magnetic resonance (NMR) data were collected on a Jeol JNM ECP-400 spectrometer (Jeol Ltd., Tokyo, Japan) using a mixture of D_2_O (with 0.01 mg/mL sodium trimethylsilylpropanesulfonate (DSS))-acetone-*d*_6_ (6:1 *v*/*v*) or dimethyl sulfoxide (DMSO)-*d*_6_ as solvents. The internal reference or residual solvent signals were utilized for referencing (D_2_O (with DSS)-acetone-*d*_6_: dH 0.00/dC 29.8, DMSO-*d*_6_: dH 2.50/dC 39.5). The peak-splitting patterns were abbreviated as m, s, d, t, dd, td, and ddd for multiplet, singlet, doublet, triplet, doublet of doublets, triplet of doublets, and doublet of doublets of doublets, respectively. Microplate (Thermo Scientific Inc., San Diego, CA, USA) and multimode plate readers (MultistkanTM GO, Thermo Scientific, Waltham, MA, USA) were used for absorbance analyses.

### 4.2. Synthesis and Characterization

#### 4.2.1. General Method for the Synthesis of 5-(*tert*-Butoxy)-4-((*tert*-butoxycarbonyl)amino)-5-oxopentanoic Acid Hydrazine Analogues

The starting material (*S*)-5-(*tert*-butoxy)-4-((*tert*-butoxycarbonyl)amino)-5-oxopentanoic acid was added to a 500 mL round-bottom flask equipped with a magnetic stir bar. Then, DCM (150 mL) was added to dissolve the starting material, and the temperature was lowered to −5 °C. Once the temperature stabilized, TEA (1.5 eq) was slowly added to the reaction mixture and stirred for approximately 10 min. Subsequently, ECF (1.5 eq) was added dropwise to the mixture over 1 h. The reaction mixture was maintained at −5 °C and stirred for an additional 4 h. In a separate 100 mL pear-shaped flask, hydrazine HCl salt with a chloride functional group was dissolved in DCM; subsequently, TEA (1.5 eq) was added. Then, this solution was slowly added to the reaction mixture and incubated for approximately 10 min. After the addition of hydrazine, the temperature was increased to room temperature (approximately 24 °C), and the reaction mixture was stirred for 16 h. After the reaction was completed, the organic phase was washed with distilled water, 1 N HCl, 0.5 M NaHCO_3_, and distilled water again to separate the layers. Subsequently, the organic phase was collected. The organic phase was dried over MgSO_4_ and concentrated using a rotary evaporator. Purification was achieved by MPLC (C18 resin) in water and MeOH.

#### 4.2.2. General Method for the Synthesis of Ramalin Chloride Derivatives

A 500 mL round-bottom flask equipped with a magnetic stir bar was charged with an appropriate Boc-glu-Hyd analog (7.0 mmol). The mixture was dissolved in 1 M HCl in EA (100 mL, 100 mmol). Then, the reaction was allowed to proceed at room temperature (24 °C) for approximately 18 h. The resulting white solid was filtered and washed with EA and *n*-hexane. The filtered white solid was dried under vacuum to obtain the Ramalin chloride derivative. The analysis results of derivatives are inchlded in the Appendix A.

*N*^5^-((2-chlorophenyl)amino)-*L*-glutamine (RA-2Cl). From (*S*)-5-(*tert*-butoxy)-4-((*tert*-butoxycarbonyl)amino)-5-oxopentanoic acid; 1.68 g, 78%, white solid; ^1^H NMR (400 MHz, DMSO-*d*_6_): *δ* 7.26 (dd, *J* = 8.4, 1.2 Hz, 1H, PhH), 7.14 (t, *J* = 7.2 Hz, 1H, PhH), 6.77 (dd, *J* = 8.4, 1.6 Hz, 1H, PhH), 6.73 (ddd, *J* = 8.4, 7.2, 1.2 Hz, 1H, PhH), 3.26 (t, *J* = 6.4 Hz, 1H, H-2), 2.37 (m, 2H, H-4), 1.92 (m, 2H, H-3); ^13^C NMR (100 MHz DMSO-*d*_6_): *δ* 171.7, 169.7, 144.7, 129.1, 127.7, 119.3, 117.1, 113.3, 53.6, 29.9, 27.0; HRESIMS *m/z* 272.0802 [M + H]^+^ (calcd for C_11_H_15_ClN_3_O_3_, 272.0800).

*N*^5^-((3-chlorophenyl)amino)-*L*-glutamine (RA-3Cl). From (*S*)-5-(*tert*-butoxy)-4-((*tert*-butoxycarbonyl)amino)-5-oxopentanoic acid; 1.73 g, 80%, brown solid; ^1^H NMR (400 MHz, DMSO-*d*_6_): *δ* 7.13 (dd, *J* = 8.2, 7.8 Hz, 1H, PhH), 6.70 (ddd, *J* = 7.8, 1.8, 0.9 Hz, 1H, PhH), 6.67 (dd, *J* = 2.3, 1.8 Hz, 1H, PhH), 6.64 (ddd, *J* = 8.2, 2.3, 0.9 Hz, 1H, PhH), 3.90 (q, *J* = 11.5, 6.0 Hz, 1H, H-2), 2.47 (m, 2H, H-4), 2.06 (m, 2H, H-3); ^13^C NMR (100 MHz, DMSO-*d*_6_): *δ* 170.8, 170.6, 150.9, 133.5, 130.4, 117.8, 111.3, 110.7, 51.5, 28.8, 25.8; HRESIMS *m*/*z* 272.0800 [M + H]^+^ (calcd for C_11_H_15_ClN_3_O_3_, 272.0800).

*N*^5^-((4-chlorophenyl)amino)-*L*-glutamine (RA-4Cl). From (*S*)-5-(*tert*-butoxy)-4-((*tert*-butoxycarbonyl)amino)-5-oxopentanoic acid; 1.79 g, 83%, pink solid; ^1^H NMR (400 MHz, DMSO-*d*_6_): *δ* 7.14 (dd, *J* = 6.8, 2.4 Hz, 2H, PhH), 6.70 (dd, *J* = 6.8, 1.6 Hz, 2H, PhH), 3.26 (t, *J* = 6.4 Hz, 1H, H-2), 2.35 (m, 2H, H-4), 1.92 (m, 2H, H-3); ^13^C NMR (100 MHz, DMSO-*d*_6_): *δ* 171.8, 169.9, 148.4, 128.4, 121.6, 113.6, 53.6, 29.8, 27.0; HRESIMS *m*/*z* 272.0800 [M + H]^+^ (calcd for C_11_H_15_ClN_3_O_3_, 272.0800).

*N*^5^-((2,3-dichlorophenyl)amino)-*L*-glutamine (RA-23Cl). From (*S*)-5-(*tert*-butoxy)-4-((*tert*-butoxycarbonyl)amino)-5-oxopentanoic acid; 1.99 g, 83%, white solid; ^1^H NMR (400 MHz, Acetone-*d*_6_/D_2_O (1:6)): *δ* 7.06 (t, *J* = 8.0 Hz, 1H, PhH), 6.92 (dd, *J* = 8.0, 1.6 Hz, 1H, PhH), 6.72 (dd, *J* = 8.0, 1.6 Hz, 1H, PhH), 4.00 (t, *J* = 6.4 Hz, 1H, H-2), 2.54 (m, 2H, H-4), 2.16 (m, 2H, H-3); ^13^C NMR (100 MHz, DMSO-*d*_6_): *δ* 171.7, 169.7, 144.7, 129.1, 127.7, 119.3, 117.1, 113.3, 53.6, 29.9, 27.0; HRESIMS *m*/*z* 306.4013 [M + H]^+^ (calcd for C_11_H_14_Cl_2_N_3_O_3_, 306.0412).

*N*^5^-((2,4-dichlorophenyl)amino)-*L*-glutamine (RA-24Cl). From (*S*)-5-(*tert*-butoxy)-4-((*tert*-butoxycarbonyl)amino)-5-oxopentanoic acid; 1.92 g, 80%, pink solid; ^1^H NMR (400 MHz, DMSO-*d*_6_): *δ* 7.06 (t, *J* = 8.0 Hz, 1H, PhH), 6.92 (dd, *J* = 8.0, 1.6 Hz, 1H, PhH), 6.72 (dd, *J* = 8.0, 1.6 Hz, 1H, PhH), 4.00 (t, *J* = 6.4 Hz, 1H, H-2), 2.54 (m, 2H, H-4), 2.16 (m, 2H, H-3); ^13^C NMR (100 MHz, DMSO-*d*_6_): *δ* 171.7, 169.7, 144.7, 129.1, 127.7, 119.3, 117.1, 113.3, 53.6, 29.9, 27.0; HRESIMS *m*/*z* 306.0412 [M + H]^+^ (calcd for C_11_H_14_Cl_2_N_3_O_3_, 306.0412).

*N*^5^-((2,6-dichlorophenyl)amino)-*L*-glutamine (RA-26Cl). From (*S*)-5-(*tert*-butoxy)-4-((*tert*-butoxycarbonyl)amino)-5-oxopentanoic acid; 2.04 g, 85%, white solid; ^1^H NMR (400 MHz, Acetone-*d*_6_/D_2_O (1:6)): *δ* 7.22 (d, *J* = 8.0, 2.0 Hz, 2H, PhH), 6.89 (t, *J* = 8.0 Hz, 1H, PhH), 3.94 (t, *J* = 6.4 Hz, 1H, H-2), 2.41 (m, 2H, H-4), 2.10 (m, 2H, H-3); ^13^C NMR (100 MHz, DMSO-*d*_6_): *δ* 173.2, 171.7, 140.3, 129.5, 125.5, 124.6, 52.7, 29.2, 25.8; HRESIMS *m*/*z* 306.0411 [M + H]^+^ (calcd for C_11_H_14_Cl_2_N_3_O_3_, 306.0412).

*N*^5^-((3,4-dichlorophenyl)amino)-*L*-glutamine (RA-34Cl). From (*S*)-5-(*tert*-butoxy)-4-((*tert*-butoxycarbonyl)amino)-5-oxopentanoic acid; 1.85 g, 77%, white solid; ^1^H NMR (400 MHz, Acetone-*d*_6_/D_2_O (1:6)): *δ* 7.20 (d, *J* = 8.4 Hz, 2H, PhH), 6.85 (d, *J* = 2.8 Hz, 1H, PhH), 6.60 (d, *J* = 8.4, 2.8 Hz, 1H, PhH), 4.03 (t, *J* = 6.4 Hz, 1H, H-2), 2.53 (m, 2H, H-4), 2.19 (m, 2H, H-3); ^13^C NMR (100 MHz, DMSO-*d*_6_): *δ* 173.9, 171.5, 148.6, 132.4, 131.2, 122.1, 114.5, 113.5, 52.7, 29.5, 25.9; HRESIMS *m*/*z* 306.0410 [M + H]^+^ (calcd for C_11_H_14_Cl_2_N_3_O_3_, 306.0412).

*N*^5^-((3,5-dichlorophenyl)amino)-*L*-glutamine (RA-35Cl). From (*S*)-5-(*tert*-butoxy)-4-((*tert*-butoxycarbonyl)amino)-5-oxopentanoic acid; 1.87 g, 78%, white solid; ^1^H NMR (400 MHz, Acetone-*d*_6_/D_2_O (1:6)): *δ* 7.20 (dd, *J* = 2.0, 1.6 Hz, 2H, PhH), 6.50 (dd, *J* = 2.8, 1.6 Hz, 2H, PhH), 4.00 (t, *J* = 6.4 Hz, 1H, H-2), 2.50 (m, 2H, H-4), 2.16 (m, 2H, H-3); ^13^C NMR (100 MHz, DMSO-*d*_6_): *δ* 174.2, 171.6, 150.5, 135.5, 119.9, 111.5, 52.7, 29.4, 25.7; HRESIMS *m*/*z* 306.0410 [M + H]^+^ (calcd for C_11_H_14_Cl_2_N_3_O_3_, 306.0412).

### 4.3. Antioxidant Activity Assay

To evaluate the antioxidant activities of Ramalin and its chloride derivatives, their radical scavenging activities were measured using DPPH. Briefly, 150 µL of Ramalin, its derivatives, and BHA solutions with concentrations of 10, 5, 2.5, and 1 mM in MeOH were mixed with 50 µL of 0.1 mM DPPH in MeOH and placed in the dark for 30 min at room temperature. The absorbances of the mixtures were measured at 540 nm.

### 4.4. Cytotoxicity and Anti-Inflammation Activity Assays

#### 4.4.1. Cell Culture

The RAW 264.7 cell line, which resembled macrophages (KCLB number 40071; Korean Cell Line Bank, Seoul, Republic of Korea), was maintained in Dulbecco’s Modified Eagle’s Medium (DMEM, Sigma-Aldrich, St. Louis, MO, USA) with the addition of a 10% heat-inactivated fetal bovine serum (FBS, Invitrogen, Burlington, ON, Canada) and 1% (*w*/*v*) antibiotic–antimycotic solution (Invitrogen, Grand Island, NY, USA) at 37 °C in a 5% CO_2_ atmosphere.

#### 4.4.2. Cytotoxicity Assay

Cell cytotoxicity was assessed using the 2,5-diphenyl-2*H*-tetrazolium bromide (MTT) colorimetric assay (3-(4,5)-dimethylthiazol-2-yl-2,5-diphenyltetrazolium bromide, Amresco, Solon, OH, USA). RAW 264.7 cells were plated at a density of 2 × 10^5^ cells/mL in 96-well plates and exposed to various concentrations of Ramalin and its derivatives for 24 h. After the incubation period, 5 µL of the MTT solution (5 mg/mL in PBS) was added to each well and incubated for an additional 4 h at 37 °C. Subsequently, 100 µL of fresh DMSO was added to dissolve the formazan crystals for 10 min. The absorbances were measured at 570 nm using a microplate reader (Thermo Scientific Inc., San Diego, CA, USA). Relative cell viabilities were determined by comparing the absorbance values of the treated samples to that of the untreated control group. All experiments were conducted in triplicate.

#### 4.4.3. Determination of NO Production

Nitrite accumulation was used as an indicator of NO production in the medium, and nitrite levels were determined by assaying the culture supernatants for nitrite using Griess reagent, which consisted of 1% sulfanilamide, 0.1% *N*-(1-naphathyl)-ethylenediamine dihydrochloride, and 5% phosphoric acid. To measure the amount of nitrite, 1 × 10^6^ cells/mL were seeded in 96-well plates and treated with the indicated concentrations of Ramalin and its derivatives at 37 °C for 1 h, followed by stimulation with 0.5 µg/mL LPS (Sigma-Aldrich, CA, USA) for 24 h in a final volume of 200 µL. Then, 100 µL of cell culture supernatants was mixed with 100 µL of Griess reagent in a 96-well plate. Sodium nitrite was used to generate a standard curve. The nitrite concentration was determined by measuring the absorbance at 540 nm using a microplate reader. All measurements were performed in triplicate.

### 4.5. Western Blot Analysis of the Anti-Inflammation and BACE-1 Inhibition Activities

#### 4.5.1. Treatment of N2a and SY-SY5Y Cells with Ramalin Chloride Derivatives

Cells were incubated with 10 μM of Ramalin and its chloride derivatives for 24 h. Then, the cells were exposed to 100 μM of hydrogen peroxide (Sigma-Aldrich) for 6 h (N2a) or 200 μM of hydrogen peroxide for 2 h (SH-SY5Y) to induce BACE-1 expression.

#### 4.5.2. Treatment of BV-2 Cells with Ramalin Chloride Derivatives

Cells were treated with 10 μM of Ramalin and its chloride derivatives for 24 h and incubated with 200 ng/mL LPS (Sigma-Aldrich) for 6 h to induce inflammation.

#### 4.5.3. Western Blot Analysis

Cells were lysed in a tissue protein extraction reagent (T-PER™, Thermo Scientific) supplemented with a protease and phosphatase inhibitor cocktail (Thermo Scientific) and incubated for 10 min on ice. Then, the lysates were centrifuged at 13,000 rpm for 10 min at 4 °C, and the resulting supernatant was used for Western blot analysis. The amount of protein was quantified using the Pierce™ BCA Protein Assay Kit (Thermo Scientific). Equal amounts of protein samples were loaded and separated by sodium dodecyl sulfate–polyacrylamide gel electrophoresis and transferred onto polyvinylidene difluoride membranes (Millipore, Darmstadt, Germany). The membranes were blocked in 5% nonfat skim milk for 1 h at room temperature before being incubated with primary antibodies against NLRP3 (Cell Signaling Technology, Danvers, MA, USA), COX-2 (Abcam, Cambridge, UK), BACE-1 (Cell Signaling Technology), and β-actin (Sigma-Aldrich) overnight at 4 °C. The membranes were washed with a 1X Tris-buffered saline with 0.1% Tween^®^ 20 detergent buffer and incubated with horseradish peroxidase-conjugated antimouse or antirabbit secondary antibodies (Millipore) for 1 h at room temperature (24 °C). Protein bands were detected using an enhanced chemiluminescence solution (Cytiva, Marlborough, MA, USA). Densitometric quantification of the protein bands was performed using ImageJ 1.54f (NIH, Bethesda, MD, USA).

#### 4.5.4. Statistical Analysis

Graphs were created and statistical analyses were performed using the GraphPad Prism 8 software. Data were analyzed using a one-way analysis of variance followed by Dunnett’s multiple comparison test. All data were expressed as the mean ± standard deviation of three independent experiments. The statistical significance was set at *p* < 0.05.

### 4.6. Tau Inhibition Activity Assay

#### 4.6.1. Tissue Culture of Adherent Cells

SH-SY5Y cells were cultured in complete growth medium: DMEM supplemented with 10% FBS, and 1% penicillin–streptomycin. Adherent SH-SY5Y cells were seeded in a 96-well plate at a density of 2 × 10^6^ cells per well for treatment. Levosimendan (Sigma, USA) was used as an inhibitor to confirm the tau inhibition effect. After treatment with 20 μM of each substance, culture was performed for 24 h. Subsequently, the supernatant was recovered and used for the ELISA assay.

#### 4.6.2. Tau ELISA Assay

Tau ELISA assays were conducted using a Human Tau ELISA kit (Abcam, Cambridge, UK). After treatment, the supernatant was diluted five-fold. The diluted supernatant, capture antibody, and detection antibody were mixed in a 2:1:1 ratio and incubated for 1 h. Subsequently, each well was washed three times with a wash buffer. After completely removing the wash buffer, 100 µL of a 3,3′,5,5′-tetramethylbenzidine developer was treated for 10 min, after which 100 µL of a stop solution was added, and the optical density was measured at 450 nm. The data were graphed by calculating the mean and standard deviation using the Prism (GraphPad, San Diego, CA, USA) program.

#### 4.6.3. Statistical Analysis

Graphs were created and statistical analyses were performed using the GraphPad Prism 8 software. Data were analyzed using a one-way analysis of variance followed by Dunnett’s multiple comparison test. All data were expressed as the mean ± standard deviation of three independent experiments. The statistical significance was set at * *p* < 0.05, ** *p* < 0.01, *** *p* < 0.005.

## 5. Conclusions

In conclusion, this study highlighted the potential of chloride-substituted Ramalin derivatives as multifunctional therapeutic agents for AD. Our results indicate that these derivatives can effectively inhibit tau aggregation and, in some cases, reduce BACE-1 expression, thereby addressing two critical pathological features of AD. The synthetic challenges encountered during the production of these derivatives were mitigated by modifying the reaction conditions, leading to the successful isolation of the functional compounds. Although BBB permeability remains a challenge, strategies such as prodrug development and nanoparticle delivery systems offer potential solutions. Notably, derivatives, such as RA-34Cl and RA-35Cl, showed the most promise, demonstrating high tau inhibition and a dual activity against BACE-1. Further studies, including structure–activity relationship analyses and in vivo evaluations using AD models, are essential for optimizing these compounds for therapeutic use. These findings underscore the importance of structural modifications in enhancing the efficacy of Ramalin derivatives, while also opening new avenues for developing effective treatments for neurodegenerative disorders such as AD.

## Data Availability

The data presented in this study are available in this article.

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
