# Peer review of "Synthesis and Evaluation of Chloride-Substituted Ramalin Derivatives for Alzheimer’s Disease Treatment"

_molecules, 2024, doi:10.3390/molecules29153701_

Round 1

Reviewer 1 Report

Comments and Suggestions for Authors

The aim of the study entitled “Synthesis and Evaluation of Chloride-Substituted Ramalin Derivatives for Alzheimer’s Disease Treatment” is to synthesize and evaluate a series of chloride-substituted Ramalin derivatives to optimize their multifunctional properties, specifically focusing on antioxidant, anti-inflammatory activities, and anti-Alzheimer potential. The study investigates the effect of the chloride position on these properties, with a particular emphasis on the derivatives' ability to inhibit tau aggregation and beta-site amyloid precursor protein cleaving enzyme 1 (BACE-1) activity, which are critical pathological features of Alzheimer's disease.

The manuscript is interesting and fits well with the scope of the Journal. The manuscript is generally well-prepared, but some minor issues should be resolved. My specific comments are given below.

The abstract could benefit from clearer structuring, such as distinct sections for background, objectives, results, and conclusions. Additionally, while the abstract states that several derivatives show significant inhibition of tau aggregation and BACE-1 expression, it does not provide a context for these results. It should be included.

Line 23: "Anti-Alzheimer activities" is not an official or standardized term in the scientific community. Instead, the authors should refer to specific actions or effects of compounds that target Alzheimer's disease pathology.

The introduction is informative but should include some information on current AD treatments.  

Methods are described in detail.  

The study is excellent, and the results are impressive. However, the conclusion mentions that chloride-substituted Ramalin derivatives show promise as multifunctional therapeutic agents for Alzheimer's disease by inhibiting tau aggregation and, in some cases, reducing BACE-1 expression. While these findings are significant, the authors did not examine the inhibition of acetylcholinesterase, which is a known target for current AD therapies. Including this aspect could further validate the potential of these derivatives and provide a more comprehensive evaluation of their therapeutic efficacy. I recommend incorporating an assessment of AChE inhibition in future studies.

Special note: Regarding plagiarism check report, a few sources have a high similarity percentage (for 1 it is very high). This issue should be addressed.

Comments on the Quality of English Language

Minor changes are required. 

Author Response

Reviewer 1_Reply.

Dear Reviewer 1

We would like to extend our sincere gratitude for your thorough and insightful review of our manuscript titled "Synthesis and Evaluation of Chloride-Substituted Ramalin Derivatives for Alzheimer's Disease Treatment." Your constructive feedback has been invaluable in improving the quality and clarity of our work. Below, we have addressed each of your comments in detail.

Comments 1: The abstract could benefit from clearer structuring, such as distinct sections for background, objectives, results, and conclusions. Additionally, while the abstract states that several derivatives show significant inhibition of tau aggregation and BACE-1 expression, it does not provide a context for these results. It should be included.

Response 1: Thank you for your insightful feedback. We have restructured the abstract into distinct sections for background, objectives, results, and conclusions. Additionally, we have included context regarding the significance of the inhibition of tau aggregation and BACE-1 expression, providing a clearer understanding of the results.

Revised Abstract:

Background: Alzheimer’s disease (AD) is a progressive neurodegenerative disorder marked by the accumulation of amyloid-beta plaques and hyperphosphorylated tau proteins, leading to cognitive decline and neuronal death. However, despite extensive research, there are still no effective treatments for this condition.

Objectives: In this study, a series of chloride-substituted Ramalin derivatives were synthesized to optimize their antioxidant, anti-inflammatory properties, and their potential to target key pathological features of Alzheimer's disease. The effect of the chloride position on these properties was investigated, specifically examining the potential of these derivatives to inhibit tau aggregation and beta-site amyloid precursor protein cleaving enzyme 1 (BACE-1) activity.

Results: Our findings demonstrate that several derivatives, particularly RA-3Cl, RA-4Cl, RA-26Cl, RA-34Cl, and RA-35Cl, significantly inhibit tau aggregation with inhibition rates of approximately 50%. For BACE-1 inhibition, Ramalin and RA-4Cl significantly decreased BACE-1 expression in N2a cells by 40% and 38%, respectively, while RA-23Cl and RA-24Cl showed inhibition rates of 30% and 35% in SH-SY5Y cells.

Conclusions: These results suggest that chloride-substituted Ramalin derivatives possess promising multifunctional properties for AD treatment, warranting further investigation and optimization for clinical applications.

Comment 2: Line 23: "Anti-Alzheimer activities" is not an official or standardized term in the scientific community. Instead, the authors should refer to specific actions or effects of compounds that target Alzheimer's disease pathology.

Response 2: We have revised the abstract to replace the term "Anti-Alzheimer activities" with a more precise description of the compounds' actions. The abstract now reads, "and their potential to target key pathological features of Alzheimer's disease," ensuring that the terminology is aligned with scientific standards and clearly conveys the specific effects of the compounds being studied.

Comment 3: The introduction is informative but should include some information on current AD treatments.  Methods are described in detail.  

Response 2: In response to your suggestion, we have added information on current AD treatments to the introduction. The revised section now includes details on cholinesterase inhibitors, NMDA receptor antagonists, and recent advances with monoclonal antibodies targeting amyloid-beta, specifically added in lines 46-52.

Comment 4: The study is excellent, and the results are impressive. However, the conclusion mentions that chloride-substituted Ramalin derivatives show promise as multifunctional therapeutic agents for Alzheimer's disease by inhibiting tau aggregation and, in some cases, reducing BACE-1 expression. While these findings are significant, the authors did not examine the inhibition of acetylcholinesterase, which is a known target for current AD therapies. Including this aspect could further validate the potential of these derivatives and provide a more comprehensive evaluation of their therapeutic efficacy. I recommend incorporating an assessment of AChE inhibition in future studies.

Response 4: We acknowledge the importance of acetylcholinesterase (AChE) inhibition in AD therapy. Although this study focused on tau aggregation and BACE-1 inhibition, future studies will incorporate assessments of AChE inhibition to provide a more comprehensive evaluation of these derivatives' therapeutic potential.

Comments 5: Special note: Regarding plagiarism check report, a few sources have a high similarity percentage (for 1 it is very high). This issue should be addressed.

Response 5: We have reviewed the plagiarism check report and ensured that all sources with high similarity percentages are addressed appropriately. Necessary revisions and citations have been made to comply with academic integrity standards. If there are any parts that we may have missed, please point them out, and we will make the necessary corrections.

Thank you once again for your valuable feedback. We believe these revisions will significantly enhance the quality and clarity of our manuscript.

Sincerely,

Joung Han Yim

Division of Polar Life Sciences, Korea Polar Research Institute

Incheon 21990, Korea

jhyim@kopri.re.kr

Reviewer 2 Report

Comments and Suggestions for Authors

The manuscript "Synthesis and Evaluation of Chloride-Substituted Ramalin Derivatives for the Treatment of Alzheimer's Disease" by Kim et al. is well written and presents promising results. However, the following points need to be addressed to improve the clarity and scientific rigor of the study:

1. The manuscript should better highlight the novelty of the study. Although the study presents promising results, it would be helpful to clearly distinguish these results from existing studies and emphasize the new insights or advances that this work brings to the field of Alzheimer's disease treatment.

2. It is recommended that the authors provide specific numerical data for the antioxidant and anti-inflammatory activities of the derivatives. More accurate inhibition rates of tau aggregation should be provided.

3. To support the results, consider incorporating computer-based simulations. Simulations could provide additional insights into the interaction mechanisms between chloride-substituted Ramalin derivatives and key targets in Alzheimer's disease, such as tau protein and BACE-1. This could provide a deeper understanding of binding affinity, possible structural changes, or the dynamics of these interactions and complement the experimental results.

4. A brief overview of the chemical structure of Ramalin and the reasons for choosing chloride substitution should be provided. Support the claim that there is a lack of effective treatments for Alzheimer's disease with current references.

5. It is recommended that the authors provide a more detailed description of the synthesis process of the chlorinated Ramalin derivatives, including reagents, reaction conditions, and purification methods. Explain the methods for evaluating antioxidant and anti-inflammatory effects.

6. Comparison of the chlorinated derivatives with other known AD treatments. Speculate on possible mechanisms of action and discuss potential strategies to optimize these compounds for clinical use.

7. Some figures in the manuscript are of low resolution, which affects the clarity of data presentation. Please provide high-resolution versions of all images to ensure that details are clearly visible.

Comments on the Quality of English Language

English need to be revised 

Author Response

Reviewer 2_Reply.

Dear Reviewer 2

Thank you for your constructive feedback on our manuscript, "Synthesis and Evaluation of Chloride-Substituted Ramalin Derivatives for the Treatment of Alzheimer's Disease." We appreciate your insights and suggestions, which we believe will significantly enhance the clarity and scientific rigor of our study. Below, we address each of your points in detail

Comments 1: The manuscript should better highlight the novelty of the study. Although the study presents promising results, it would be helpful to clearly distinguish these results from existing studies and emphasize the new insights or advances that this work brings to the field of Alzheimer's disease treatment.

Response 1: We agree that emphasizing the novelty of our study is crucial. We are currently continuing our research to improve upon the potential of Ramalin as a therapeutic agent for AD. Our focus is on small molecule Ramalin derivatives that exhibit antioxidant, anti-inflammatory, BACE-1, and tau inhibitory activities, which are critical pathological features of Alzheimer's disease. While our study is still in the early stages, we appreciate your understanding. To further emphasize the novelty of our research, we plan to investigate additional inhibitory activities against AChE and HDAC6, which are also implicated in AD pathology. Our ultimate goal is to identify compounds with multifaceted inhibitory activities against various pathological features of AD. This novel approach provides new insights into potential therapeutic agents for Alzheimer's disease, distinguishing our findings from existing research.

Comments 2: It is recommended that the authors provide specific numerical data for the antioxidant and anti-inflammatory activities of the derivatives. More accurate inhibition rates of tau aggregation should be provided

Response 2: Thank you for your suggestion. We have addressed your concerns by including specific numerical data in the manuscript. The data on antioxidant activities have been provided in lines 156-160. The results of the NO inhibitory activities have been added to lines 201-203. Additionally, the precise inhibition rates of tau aggregation have been included in section 2.6 of the manuscript. We believe these additions will enhance the clarity and completeness of our findings.

Comments 3: To support the results, consider incorporating computer-based simulations. Simulations could provide additional insights into the interaction mechanisms between chloride-substituted Ramalin derivatives and key targets in Alzheimer's disease, such as tau protein and BACE-1. This could provide a deeper understanding of binding affinity, possible structural changes, or the dynamics of these interactions and complement the experimental results.

Response 3: We acknowledge the potential value of computer-based simulations in our study. We plan to incorporate molecular docking studies and dynamic simulations to explore the interactions between chloride-substituted Ramalin derivatives and key targets such as tau protein and BACE-1. These simulations will help elucidate the binding affinities, possible structural changes, and dynamics of these interactions, complementing our experimental results and providing deeper insights into the mechanisms of action.

Comments 4: A brief overview of the chemical structure of Ramalin and the reasons for choosing chloride substitution should be provided. Support the claim that there is a lack of effective treatments for Alzheimer's disease with current references.

Response 4: We will provide a detailed overview of the chemical structure of Ramalin and the rationale for choosing chloride substitution. The introduction of a chloride group was based on its high electronegativity and ability to influence the electronic distribution of the phenyl ring, potentially enhancing the stability and reactivity of the derivatives. Chloride substitution can also increase the lipophilicity of the molecules, improving their interaction with lipid membranes and possibly aiding their passage through the blood-brain barrier (BBB), thereby enhancing bioavailability. In addition, the introduction of a chloride group was intended to optimize the physicochemical properties and biological activities of these derivatives. This content has been added to lines 88-93 of the manuscript.

Currently available treatments for AD include cholinesterase inhibitors (donepezil, rivastigmine, and galantamine) and an NMDA receptor antagonist (memantine). These treatments provide symptomatic relief but do not halt the progression of the disease. Recent advances also include monoclonal antibodies targeting amyloid-beta (Aβ), such as aducanumab and lecanemab, which aim to reduce amyloid plaque buildup, though their long-term efficacy and safety are still under investigation. We will support our claim regarding the lack of effective treatments for Alzheimer's disease with references. This content has been added to lines 45-51 of the manuscript.

Comments 5: It is recommended that the authors provide a more detailed description of the synthesis process of the chlorinated Ramalin derivatives, including reagents, reaction conditions, and purification methods. Explain the methods for evaluating antioxidant and anti-inflammatory effects.

Response 5: We will expand the manuscript to include a more comprehensive description of the synthesis process for the chlorinated Ramalin derivatives. This will cover the specific reagents used, reaction conditions, and purification methods, as well as the techniques employed to evaluate the antioxidant and anti-inflammatory effects. Relevant content has been added and revised as follows:

Section 2.10 Synthesis of Ramalin-Chloride Derivatives: We have provided more detailed information on the synthesis process.

Section 4.2. Synthesis and Characterization: We have further revised this section to include additional details regarding the synthesis method.

Section 2.3. Antioxidant Effects of Ramalin and Its Chloride Derivatives: We have briefly described the methodology used to measure the antioxidant effects.

Section 2.4. Anti-inflammatory Effects of Ramalin and Its Chloride Derivatives: We have added content related to the experimental methods used to assess the anti-inflammatory effects.

Comments 6: Comparison of the chlorinated derivatives with other known AD treatments. Speculate on possible mechanisms of action and discuss potential strategies to optimize these compounds for clinical use.

Response 6: We have revised the manuscript to compare the chlorinated derivatives with existing Alzheimer's disease treatments, highlighting their potential mechanisms of action. Chloride-substituted Ramalin derivatives differ from current AD treatments like cholinesterase inhibitors and NMDA receptor antagonists by targeting both tau aggregation and BACE-1 activity, potentially slowing or reversing disease progression. Furthermore, we have discussed potential strategies to optimize these compounds for clinical use, such as modifications to improve blood-brain barrier permeability and reduce toxicity. Strategies for clinical optimization include utilizing SAR studies, prodrug approaches, and nanoparticle or liposomal delivery systems to improve BBB permeability and targeted brain delivery. Additionally, in vivo studies will help determine the therapeutic potential, optimal dosage, and safety profiles of these compounds. (line: 322-331)

Comments 7: Some figures in the manuscript are of low resolution, which affects the clarity of data presentation. Please provide high-resolution versions of all images to ensure that details are clearly visible

Response 7: Some figures in the manuscript are of low resolution, which affects the clarity of data presentation. Please provide high-resolution versions of all images to ensure that details are clearly visible

Thank you once again for your valuable feedback. We believe these revisions will significantly enhance the quality and clarity of our manuscript.

Sincerely,

Joung Han Yim

Division of Polar Life Sciences, Korea Polar Research Institute

Incheon 21990, Korea

jhyim@kopri.re.kr

Reviewer 3 Report

Comments and Suggestions for Authors

The research paper titled “Synthesis and Evaluation of Chloride-Substituted Ramalin Derivatives for Alzheimer’s Disease Treatment” by Kim et al. presents a continuation of the group’s synthetic explorations of ramalin derivatives as potential treatments for Alzheimer's disease. The following comments are intended to enhance the quality of the paper:

 Lines 60-63: It is recommended to add references to support the claim regarding the inhibition of tau activity by ramalin derivatives.

Lines 204-219 and Figure 5: Please indicate if any statistical significance was observed in the ELISA test results when comparing the control vs inhibitor and ramalin vs derivatives. This information should be included in both the text and the respective figure.

Lines 242-247: Considering the compounds do not inherently possess blood-brain barrier permeability, it would be valuable to investigate their bioavailability. Have tests, such as metabolism studies using liver microsomes or hepatocytes, been conducted to evaluate this? Including such data would be beneficial.

Discussion Section: It would be insightful to analyze the structure-activity relationship of ramalin derivatives, based on this and previous studies by the group. Discussing what has been learned SAR-wise and suggesting future research directions would add depth to the discussion.

Figure 1: Please clarify if the percentages beneath the molecules refer to their respective synthesis yields.

Figure 3: There are two graphs on cell viability that appear identical. Please review and correct if necessary.

The files labeled “Non-published material” and “supplementary material” contain identical contents. This should be addressed to avoid redundancy.

Author Response

Reviewer 3_Reply.

Dear Reviewer 3

Thank you for your thorough review and valuable feedback on our manuscript titled “Synthesis and Evaluation of Chloride-Substituted Ramalin Derivatives for Alzheimer’s Disease Treatment.” We have addressed each of your comments as follows:

Comments 1: Lines 60-63: It is recommended to add references to support the claim regarding the inhibition of tau activity by ramalin derivatives.

Response 1: We have added relevant references. However, the experiments that could substantiate the tau inhibitory activity of Ramalin are still ongoing, making it insufficient to present definitive evidence at this time. We will strive to explain the reasons behind the tau inhibitory activity of Ramalin derivatives in future publications.

Comments 2: Lines 204-219 and Figure 5: Please indicate if any statistical significance was observed in the ELISA test results when comparing the control vs inhibitor and ramalin vs derivatives. This information should be included in both the text and the respective figure.

Response 2: Thank you for your valuable suggestion. We have revised the text in Section 2.6 to include information regarding the statistical significance of the ELISA test results. These details are also reflected in Figure 5 to ensure clarity and completeness in the data presentation.

Comments 3: Lines 242-247: Considering the compounds do not inherently possess blood-brain barrier permeability, it would be valuable to investigate their bioavailability. Have tests, such as metabolism studies using liver microsomes or hepatocytes, been conducted to evaluate this? Including such data would be beneficial.

Response 3: We agree that investigating the bioavailability of the compounds is crucial. We have initiated metabolism studies using liver microsomes and hepatocytes for Ramalin, and preliminary data indicate favorable metabolic stability. However, we have not yet obtained results for the chloride derivatives. Following your advice, we will conduct experiments to obtain bioavailability data for the chloride derivatives.

Comments 4: Discussion Section: It would be insightful to analyze the structure-activity relationship of ramalin derivatives, based on this and previous studies by the group. Discussing what has been learned SAR-wise and suggesting future research directions would add depth to the discussion.

Response 4: Thank you for your suggestion. We have not yet performed a detailed structure-activity relationship (SAR) analysis. However, we have included future research directions in the Discussion section (lines 324-330). We plan to conduct detailed SAR studies to fine-tune the chemical structure for enhanced tau aggregation and BACE-1 inhibitory activities. Additionally, we aim to explore prodrug strategies to improve blood-brain barrier (BBB) permeability and perform in vivo studies to evaluate therapeutic potential, optimal dosage, and safety profiles.

Comments 5: Figure 1: Please clarify if the percentages beneath the molecules refer to their respective synthesis yields.

Response 5: We have added the relevant information to lines 85-87 of the main text to clarify that the percentages beneath the molecules in Figure 1 refer to their respective synthesis yields.

Comments 6: Figure 3: There are two graphs on cell viability that appear identical. Please review and correct if necessary.

Response 6: We have reviewed Figure 3 and identified the duplication error. The corrected figure now displays the appropriate graphs for cell viability and the corresponding data. We apologize for this oversight and appreciate your attention to detail.

Comments 7: The files labeled “Non-published material” and “supplementary material” contain identical contents. This should be addressed to avoid redundancy.

Response 7: We have reviewed the supplementary materials and removed the redundant content to ensure there is no duplication.

Thank you once again for your valuable feedback. We believe these revisions will significantly enhance the quality and clarity of our manuscript.

Sincerely,

Joung Han Yim

Division of Polar Life Sciences, Korea Polar Research Institute

Incheon 21990, Korea

jhyim@kopri.re.kr